

# Music Assistant
## Inteligentny Asystent Muzyczny do Spotify



**Autorzy**: Krzysztof Głowacz · Mateusz Luberda · Bartosz Rodowicz · Franciszek Suszko

**Opiekun:** Bernadetta Maleszka

**Streszczenie**

Music Assistant to aplikacja mobilna zintegrowana z platformą Spotify, która redefiniuje sposób dostarczania muzyki użytkownikom poprzez spersonalizowane profile muzyczne. System eliminuje potrzebę ręcznego zarządzania playlistami automatycznie dobierając spójne rekomendacje utworów w obrębie danego profilu. Preferencje muzyczne są dynamicznie aktualizowane na podstawie aktywności użytkownika. Zaletą aplikacji jest możliwość integracji z urządzeniami z systemem Wear OS, co pozwala na sterowanie odtwarzaniem muzyki bezpośrednio z inteligentnego zegarka (ang. *smartwatch*). Zdarzenia przechwytywane przez sensory takiego urządzenia mogą być opcjonalnie przypisane do poszczególnych profili, dzięki czemu aplikacja może automatycznie sugerować dany profil w zależności od wykrytego zdarzenia.

## 1 WSTĘP

Poszukiwanie odpowiedniej piosenki na dany moment poprzez ciągłe pomijanie kolejnych utworów bywa frustrujące. Tworzenie własnych playlist jest czasochłonne i wymaga częstych aktualizacji, aby odpowiadały naszym zmieniającym się preferencjom. Dodatkowo, dostępne opcje personalizacji na platformach streamingowych opierają się głównie na utworach, które już dobrze znamy, co ogranicza możliwość odkrywania nowej muzyki. Te niedoskonałości zainspirowały nas do utworzenia aplikacji zintegrowanej ze Spotify, która nie tylko eliminuje te problemy, ale także wprowadza szeroki wachlarz opcji personalizacji. Dzięki temu Music Assistant dostosowuje się w sposób unikalny i wyjątkowy do preferencji każdego użytkownika, zapewniając jeszcze lepsze doświadczenie muzyczne.

### 1.1 Wymagania

Projekt Music Assistant powstał z myślą o użytkowniku. Chcemy zminimalizować jego udział w wyborze utworów oraz pozwolić na odkrywanie muzyki w sposób bardziej intuicyjny i spersonalizowany. W związku z tym, postawiliśmy sobie następujące wymagania:

- **Profile zamiast playlist**: Aktywność użytkownika w naszej aplikacji ogranicza się do utworzenia profilu muzycznego poprzez odpowiedź na kilka pytań. Następnie, na podstawie zebranych danych, system automatycznie dobiera utwory pasujące do profilu. Użytkownik nie ingeruje bezpośrednio w ich wybór, co przekłada się na oszczędność czasu.

- **Dynamiczna personalizacja**: Preferencje muzyczne użytkownika są dynamicznie aktualizowane na podstawie jego aktywności, co pozwala na dostarczanie spersonalizowanych rekomendacji przez cały czas działania profilu, a nie tylko po jego utworzeniu. Oceny aktywności użytkownika dzielą się na jawne - ręcznie wprowadzane oceny utworów, oraz niejawne - dane zbierane automatycznie podczas odtwarzania muzyki [1]. Te pierwsze mają większe znaczenie przy aktualizacji preferencji użytkownika.

- **Integracja z Wear OS**: Aplikacja jest zintegrowana z systemem Wear OS, co pozwala na sterowanie odtwarzaniem muzyki bezpośrednio z inteligentnego zegarka. Zbieranie danych z sensorów pozwala na automatyczne sugerowanie przełączenia profilu w zależności od wykrytego zdarzenia (np. wzrost tętna, wykrycie chodu), co minimalizuje wymagane ingerencje użytkownika.

- **Tworzenie konta**: Integracja z platformą Spotify minimalizuje czas potrzebny na rejestrację. Użytkownik loguje się za pomocą swojego konta Spotify Premium [3] i uzyskuje dostęp do wszystkich funkcji aplikacji.

W celu zrealizowania powyższych wymagań zdefiniowaliśmy przypadki użycia dla wszystkich funkcjonalności aplikacji. Wyróżniamy trzech aktorów: gościa, użytkownika oraz administratora. Gość po autoryzacji przez platformę Spotify staje się użytkownikiem. Głównym zadaniem użytkownika jest utworzenie profilu muzycznego. Aby to zrobić musi podać nazwę profilu, wybrać kolor, ikonę oraz opcjonalnie odpowiedzieć na pytania z ankiety. Jeśli używa zegarka z systemem Wear OS, może zdefiniować warunki zmiany profilu (np. wykrycie chodu), zwane dalej wyzwalaczami (ang. *trigger*). Po utworzeniu profilu może odtwarzać muzykę zgodną z jego charakterystyką. W ramach odtwarzania muzyki może oceniać utwory, przewijać je, cofać się do poprzednich oraz je zatrzymywać. Użytkownik ma również możliwość zarządzania swoimi profilami poprzez ich edycję i usuwanie. Administrator może zarządzać użytkownikami, a więc edytować czy usuwać ich profile oraz konta. Ma także dostęp do logów, które zawierają informacje o aktywności użytkowników, oraz do narzędzi monitorowania systemu, aby czuwać nad jego niezawodnością.

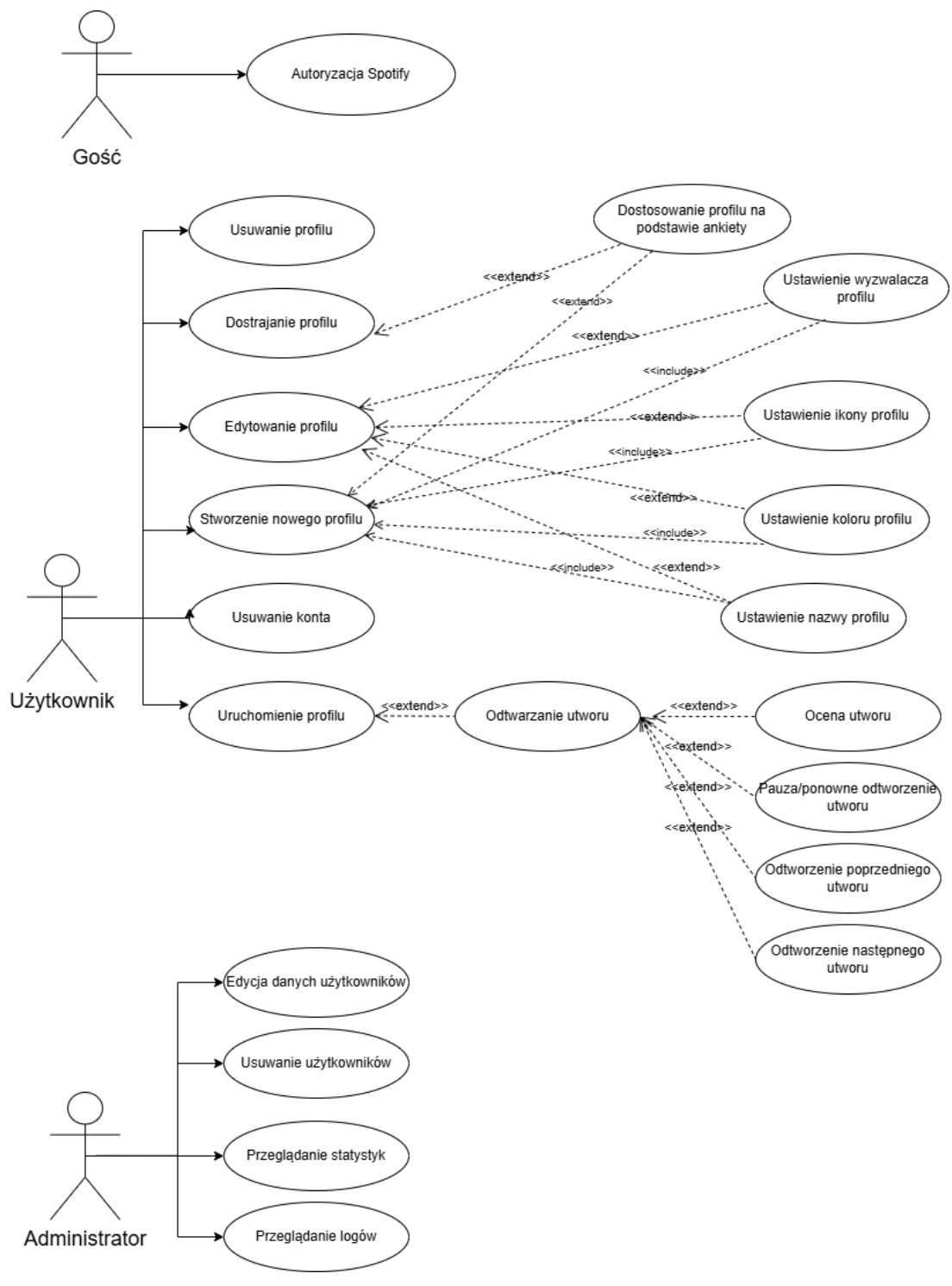

Rysunek 1: Diagram przypadków użycia

## 1.2  Cele biznesowe

- **Zwiększanie zaangażowania użytkowników**: Chcemy utworzyć aplikację, która będzie integrować się z codziennym życiem użytkownika oraz minimalizować czas spędzony przy wybieraniu muzyki. Tworzenie spersonalizowanych profili z rekomendacjami utworów oraz ich automatyczne odtwarzanie w zależności od aktywności pozwala budować silniejsze zaangażowanie.

- **Rozszerzanie grupy docelowej**: Integrowanie aplikacji z systemem Wear OS oraz wprowadzanie funkcji wykorzystujących sensory inteligentnych zegarków umożliwia dotarcie do użytkowników aktywnych fizycznie, a także osób ceniących sobie nowoczesne technologie.

- **Budowanie przewagi konkurencyjnej**: Wdrażanie zaawansowanych opcji personalizacji, odkrywanie nowych utworów oraz integracja z systemem Wear OS pozwala aplikacji wyróżniać się na tle istniejących rozwiązań. Wejście na rynek streamingu muzyki pozwala potencjalnie nawiązać współpracę z innymi platformami (np. Apple Music).

## 1.3  Cele techniczne

- **Rozwijanie systemu rekomendacji muzycznych**: Implementacja algorytmu, który dynamicznie analizuje aktywność użytkownika, dostosowuje się do jego preferencji i na bieżąco dostarcza rekomendowane utwory.

- **Integracja ze Spotify SDK i systemem Wear OS**: Synchronizowanie aplikacji ze Spotify SDK [3] pozwala zarządzać muzyką bezpośrednio z aplikacji, a wsparcie dla Wear OS dodaje funkcjonalności, takie jak automatyczne przełączanie profili na podstawie aktywności.

- **Projektowanie skalowalnego i wydajnego systemu**: Utworzenie architektury, która umożliwia łatwe dodawanie nowych funkcji, a także zapewnia wysoką wydajność i stabilność aplikacji. Nasza aplikacja jest gotowa na wdrożenie produkcyjne i może obsłużyć dużą liczbę użytkowników jednocześnie dzięki zastosowaniu technologii takich jak Apache Kafka, Docker Swarm, Celery czy FastAPI.

## 1.4  Korzyści projektu

Dzięki realizacji powyższych celów projekt Music Assistant przynosi korzyści zarówno dla użytkowników, zespołu projektowego, jak i biznesu:

- Użytkownicy zyskują narzędzie eliminujące potrzebę ręcznego zarządzania playlistami oraz nowe możliwości odkrywania muzyki. Integracja z systemem Wear OS minimalizuje czas spędzony na wybieraniu muzyki.

- Projekt stanowi doskonałą okazję do nauki i rozwój umiejętności programistycznych czy inżynierskich. Efektem końcowym prac jest gotowy produkt, który może być wdrożony w środowisku produkcyjnym. Architektura systemu pozwala na łatwe rozszerzenie aplikacji w zależności od potrzeb (np. dodanie nowych funkcji, integracja z innymi platformami streamingowymi).

- Innowacyjne podejście do rekomendacji muzycznych oraz wyróżnienie się na tle konkurencji może w przyszłości przynieść korzyści finansowe. Wysoka jakość i unikalność aplikacji otwiera drogę do nowych modeli komercjalizacji.

# 2  PRACE ZWIĄZANE Z TEMATEM

## 2.1  Analiza istniejących rozwiązań

Od dłuższego czasu najpopularniejsze na rynku muzycznym są platformy streamingowe. Wciąż jednak nie brakuje słuchaczy klasycznego radia. Coraz popularniejsze stają się radia internetowe. Wszystkie te rozwiązania mają swoje wady i zalety:

- **Platformy streamingowe**: Spotify, Apple Music, Tidal, Youtube Music, SoundCloud, Amazon Music. Wszystkie te platformy oferują dostęp do ogromnej bazy utworów, a także rekomendują słuchaczom nowe treści w postaci playlist lub w formie „radia" opartego o przesłuchany utwór. Należy jednak zwrócić uwagę na to, że większość z nich opiera się głównie na statycznych modelach rekomendacji [5], które w ograniczony sposób uwzględniają dynamiczne zmiany preferencji użytkowników oraz kontekst użytkowania.

· **Stacje radiowe**: RMF FM, Radio Zet, Radio Eska. Stacje radiowe oferują dostęp do muzyki w formie ciągłego odtwarzania, bez konieczności wybierania konkretnych utworów. Wadą tego rozwiązania jest brak personalizacji. Słuchacz może łatwo znaleźć stację odpowiadającą jego preferencjom, ale nie ma wpływu na to, jakie utwory zostaną odtworzone.

Projekt Music Assistant łączy w sobie zalety obu rozwiązań. Dzięki dynamicznemu systemowi rekomendacji muzycznych, użytkownik otrzymuje spersonalizowane propozycje utworów, które odpowiadają jego aktualnym preferencjom. Ogromną rolę w platformach streamingowych oraz radiach odgrywa popularność utworów, co sprawia, że użytkownik może nie odkryć nowych, mniej znanych artystów. Nasze rozwiązanie, które skupia się wyłącznie na numerycznej charakterystyce utworów, pozwala na odkrywanie muzyki mniej znanej, choć równie dobrej. Co więcej, dostępne na rynku produkty nie integrują technologii typu Wear OS oraz sensorów urządzeń do dynamicznego aktywowania profili muzycznych. Wprowadzenie tej funkcjonalności jest dużą zaletą Music Assistant. Chcemy, aby nasza aplikacja, dzięki integracji z inteligentnymi zegarkami, stała się prawdziwym muzycznym asystentem użytkownika.

## 2.2 Co wyróżnia Music Assistant

· **Dynamiczne profile muzyczne**: System oparty na algorytmie, który na bieżąco analizuje aktywność użytkownika i dostosowuje treści do jego aktualnych potrzeb. Stanowi to innowacyjne podejście w dziedzinie personalizacji muzyki.

· **Integracja z Wear OS i wykorzystanie sensorów**: Możliwość powiązania konkretnych zdarzeń (np. spadek tętna) wykrywanych przez sensory inteligentnego zegarka z profilami muzycznymi pozwala na dynamiczne sugestie profilu. Rozwiązanie to czyni aplikację wyjątkowo intuicyjną i oszczędza czas użytkownika.

· **Nowatorski system rekomendacji**: Implementacja systemu rekomendacji pozwala na nieustanne odkrywanie nowej muzyki w oparciu o aktualne preferencje, co zwiększa wartość aplikacji w kontekście eksploracji nowych utworów.

## 2.3 Główne założenia projektu

· **Wybór technologii**:

– **Aplikacja mobilna**: Wybraliśmy Kotlin jako język programowania oficjalnie wspierany przez Google dla systemu Android. Zapewnia nam stabilność, łatwą integrację z natywnymi funkcjami oraz wysoką wydajność. Do tworzenia interfejsu użytkownika wykorzystaliśmy Jetpack Compose, który umożliwia deklaratywne tworzenie dynamicznych i nowoczesnych interfejsów użytkownika przy minimalnym nakładzie pracy. Jego największą zaletą jest łatwa integracja z systemem Wear OS.

– **Aplikacja na inteligentny zegarek**: System Wear OS jest oficjalnym systemem operacyjnym dla urządzeń ubieralnych (ang. *wearables*) od Google. Postawiliśmy na niego ze względu na wsparcie dla zaawansowanych sensorów oraz wbudowanych algorytmów interpretacji ich odczytów. WearOS umożliwia też łatwą integrację z aplikacją mobilną.

– **Backend**: Apache Kafka to system kolejkowania wiadomości, który doskonale sprawdza się w obsłudze dużej liczby zdarzeń w czasie rzeczywistym. Dzięki temu mamy możliwość przesyłania dużej ilości informacji o aktywności użytkownika, które są następnie wykorzystywane do tworzenia nowych rekomendacji. Zdecydowaliśmy się także na użycie Celery do zarządzania zadaniami w tle. To narzędzie do obsługi kolejek zadań asynchronicznych pozwala na wykonywanie zadań w tle w sposób niezależny od żądającego. W projekcie umożliwia płynne przetwarzanie rekomendacji.

– **Integracja z API Spotify**: Spotify udostępnia kompleksowe API [4] z ogromną ilością danych o utworach i ich charakterystykach. To dzięki numerycznym charakterystykom utworów jesteśmy w stanie opracować system rekomendacji, który dostarcza użytkownikowi muzykę odpowiadającą jego preferencjom.

– **Proces wdrożeniowy**: Do procesu wdrożeniowego wybraliśmy podejście Continuous Integration and Continuous Deployment (CI/CD), ponieważ pozwala ono na szybkie i efektywne wprowadzanie zmian oraz ich testowanie. Wykorzystaliśmy narzędzie GitLab CI/CD [2], które umożliwia automatyczne budowanie, testowanie oraz wdrażanie aplikacji po każdej zmianie w repozytorium.

– **Zarządzanie kontenerami**: Wszystkie komponenty aplikacji zostały zorganizowane jako oddzielne kontenery Docker. Do ich zarządzania wykorzystaliśmy Docker Swarm, który umożliwia łatwe wdrażanie i skalowanie aplikacji w środowisku wielokontenerowym oraz wielohostowym wraz z automatycznym zarządzaniem kontenerami i ich koordynacją. Dzięki temu, mogą one działać na różnych maszynach. Rozwiązanie to gwarantuje wysoką dostępność naszej aplikacji.

– **Rozwiązania chmurowe**: Zastosowanie podejścia *Infrastructure as a Code* przy użyciu Terraform'a pozwoliło nam na łatwe wdrażanie i utrzymanie naszej platformy w chmurze. Kluczowe elementy infrastruktury to: RDS (baza danych), maszyny EC2 oraz AWS Systems Manager (używany do zarządzania konfiguracją Docker Swarm). Obok Terraform'a zastosowaliśmy także HashiCorp Vault Secrets do zarządzania tajnymi danymi, takimi jak hasła, klucze API czy tokeny dostępu. Nasza infrastruktura została zaprojektowana zgodnie z najlepszymi praktykami dotyczącymi sieci, zabezpieczeń i izolacji środowiska. Dzięki temu tylko nasz zespół oraz GitLab Runner uruchamiający CI/CD będą mogli wprowadzać zmiany w infrastrukturze naszej aplikacji.

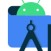 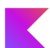 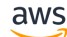 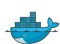 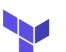 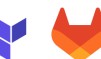 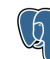 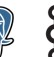 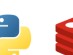 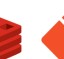 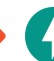 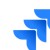 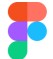

· **Ograniczenia czasowe**: Projekt należało zrealizować w 10 tygodni. Ze względu na ograniczony czas realizacji oraz wymagania związane z kursem, zdecydowaliśmy się zastosować metodykę Agile w celu elastycznego reagowania na problemy, dzieląc projekt na iteracje. Łącznie odbyliśmy 6 sprintów. Każdy miał jasno określony cel dla wszystkich członków zespołu.

– **Sprint 0** (1 tydzień): Celem zerowej iteracji było zdefiniowanie zakresu projektu oraz określenie wspólnej wizji dla wszystkich członków zespołu. Rozpoczęliśmy pracę nad dokumentacją, w tym analizą wymagań oraz projektowaniem architektury systemu. Powstały również pierwsze prototypy interfejsu i podstawy infrastruktury chmurowej.

– **Sprint 1** (2 tygodnie): W pierwszej iteracji skupiliśmy się na integracji ze Spotify API, implementacji bazy pod aplikację mobilną oraz dalszym budowaniem infrastrukutry. Zespół podzielił się na dwie grupy: jedna zajmowała się backendem, a druga frontendem. Celem sprintu było przygotowanie podstaw do dalszego rozwoju aplikacji.

– **Sprint 2** (2 tygodnie): Na drugą iterację zaplanowaliśmy implementację podstawowych funkcjonalności aplikacji mobilnej, takich jak autoryzacja użytkownika, odtwarzanie muzyki oraz zarządzanie profilami. W tym czasie zaczęliśmy również pracę nad systemem rekomendacji oraz wersją aplikacji na inteligentny zegarek.

– **Sprint 3** (2 tygodnie): Celem trzeciej iteracji było osiągniecie MVP, czyli minimalnej i akcpeptowalnej wersji produktu. Wymagało to pełnej integracji frontendu z backendem, wersji demo systemu rekomendacji oraz integracji z Wear OS.

– **Sprint 4** (2 tygodnie): W czwartej iteracji skupiliśmy się na testowaniu i poprawianiu błędów w aplikacji. Zespół pracował nad usprawnieniem interfejsu, optymalizacją kodu, ulepszaniem systemu rekomendacji oraz dodawaniem nowych funkcjonalności, które wprowadziły wartość dodaną dla projektu, lecz nie były konieczne do osiągnięcia MVP. Kontynuowaliśmy również pracę nad dokumentacją.

– **Sprint 5** (1 tydzień): Ostatni, krótszy sprint poświęciliśmy na dokończenie dokumentacji oraz implementację brakujących testów.

· **Zasoby**:

– **Pomoce techniczne**: W trakcie realizacji projektu korzystaliśmy z narzędzi takich jak GitLab, Jira, czy Figma w celu usprawnienia zarządzania kodem, zadaniami oraz określeniem wspólnej i klarownej wizji projektu.

– **Zasoby ludzkie**: Zespół składał się z 4 osób, z których każda miała przypisaną rolę. Dwie osoby zajmowały się backendem (Krzysztof Głowacz, Franciszek Suszko), a dwie frontendem (Mateusz Luberda, Bartosz Rodowicz). Wszyscy członkowie zespołu wspólnie odpowiadali za przygotowanie dokumentacji technicznej.

· **Problemy**:

– Największą trudnością okazała się być ankieta oferowana użytkownikom podczas tworzenia nowego profilu. Jej celem jest utworzenie wstępnej charakterystyki profilu, na podstawie której wygenerowane zostaną pierwsze rekomendacje. Szukając balansu między dokładnością,

intuicyjnością i prostotą zdecydowaliśmy się wprowadzić dwustopniową ankietę. W pierwszym kroku prezentowane są użytkownikowi symboliczne obrazki w formie Pixel Art, które ukazują pewne sceny z codziennego życia. Zadaniem użytkownika jest wybranie tych, z którymi utożsamia się najbardziej w kontekście danego profilu. Wybrane obrazki są następnie mapowane na wartości numeryczne (arbitralnie przypisane na podstawie przeprowadzonej ankiety środowiskowej), po czym na ich podstawie generowane są pierwsze rekomendacje muzyczne. W drugim kroku ankiety użytkownik wybiera te utwory z przedstawionych mu rekomendacji, które najbardziej odpowiadają tworzonemu profilowi. Utwory te są na końcu przetwarzane, po czym obliczane zostają bazowe parametry profilu, co kończy daną ankietę. Sam koncept interpretacji obrazków w kontekście preferencji muzycznych jest jednak zagadnieniem bardzo złożonym i mógłby być tematem osobnej pracy naukowej.

– Wielu problemów przysporzyła także implementacja wyzwalaczy na zegarku wyposażonym w system Wear OS. Kluczowym wyzwaniem było określenie metody zbierania danych z sensorów. Tryb aktywny pozwala na rejestrację danych w odstępach 0,5–1 sekundy, co zapewnia szybką reakcję na aktywność użytkownika. Niestety, znacząco zwiększa zużycie energii, co stanowi istotne ograniczenie w przypadku urządzeń typu wearable. Ostatecznie zdecydowaliśmy się na pasywny monitoring. Zapewnia on paczki danych z sensorów co około 1 min, co jest wystarczająco szybkie, aby skutecznie reagować na aktywność użytkownika. Jednocześnie nie powoduje zwiększenia częstotliwości próbkowania (ang. *sampling*) sensorów i nadaje się do długotrwałego monitorowania aktywności.

## 3  WYNIKI

Podczas prac nad implementacją projektu osiągnięte zostały główne cele biznesowe i techniczne (sekcje 1.2 i 1.3). Spełnionio wszystkie założone wymagania (sekcja 1.1). Aplikacja Music Assitant oferuje następujące funkcjonalności:

· Integracja aplikacji ze Spotify SDK oraz autoryzacja użytkownika kontem Spotify Premium.

· Tworzenie nowych profili muzycznych dwustopniową ankietą, która ma na celu zmniejszyć zimny start (ang. *cold start*) danego profilu.

· Zarządzanie profilem: aktualizacja danych, ponowne przeprowadzenie ankiety w celu poprawienia dopasowania parametrów muzycznych do preferencji użytkownika oraz usunięcie profilu.

· Integracja aplikacji z urządzeniami wyposażonymi w system Wear OS i możliwość sterowania odtwarzaniem muzyki z poziomu inteligentnego zegarka.

· Definiowanie wyzwalacza opartego o odczyty sensorów zegarka, aby umożliwić automatyczne sugerowanie uruchomienia danego profilu po wystąpieniu określonego zdarzenia.

· Generowanie rekomendacji muzycznych wyłącznie na podstawie charakterystyki profilu oraz ocen użytkownika.

Diagram kontenerów, który ukazuje architekturę systemu w ostatecznym kształcie, został przedstawiony na rys. 2.

Music Assistant spełnia postawione wymagania funkcjonalne: użytkownik może w prosty sposób skonfigurować swoje konto, dodać nowe profile, powiązać je z wyzwalaczem w systemie Wear OS, a następnie odtwarzać muzykę bez dodawania utworów do kolejki. Identyfikacja użytkownika opiera się na jego unikalnym identyfikatorze nadawanym przez Spotify, dzięki czemu możliwe jest odtwarzanie skonfigurowanych profili muzycznych z dowolnego smartfona. W celu ochrony prywatności użytkownika w całej komunikacji aplikacji mobilnej z API, a także w samej bazie danych, identyfikator ten pamiętany jest wyłącznie w postaci haszu (ang. *hash*), przez co niemożliwe jest późniejsze powiązanie danych o aktywności użytkownika Music Assistant z kontem Spotify.

Idea stojąca za Music Assistant opiera się w całości na numerycznej charakterystyce utworów dostarczanej przez Spotify API. Każdy utwór opisany jest przez zestaw wartości: akustyczność, taneczność, energia, instrumentalność, żywotność, głośność, mowa, tempo, pozytywność. Rolą systemu rekomendacji Music Assistant jest odpowiednie dobranie kombinacji tych wartości, aby pobrać ze Spotify API utwory, które spełniają zadane warunki (z pewną ustaloną dokładnością). W czasie działania systemu kombinacje te (zwane charakterystykami profilu muzycznego) dynamicznie się zmieniają analizując jawne i niejawne oceny użytkownika. Na podstawie zebranych ocen przesłuchane utwory klasyfikowane są jako pozytywne, negatywne lub neutralne, po czym następuje aktualizacja parametrów profilu, która ma na celu zbliżyć profil pod względem charakterystyki do utworów pozytywnych, jednocześnie oddalając go

od utworów negatywnych. Dzięki ciągłym aktualizacjom każdy profil zmienia się wraz ze zmieniającymi się preferencjami użytkownika.

Należy także wspomnieć o innej zalecie, która jest konsekwencją opisanego podejścia - Music Assistant nie bierze pod uwagę popularności utworów i artystów, skupiając się wyłącznie na numerycznym opisie treści, przez co użytkownik może odkryć nowe treści, które w większości byłyby pomijane przez algorytmy klasycznych platform streamingowych ze względu na swoją niską popularność.

Pierwsze testy aplikacji napawają optymizmem - stanowią potwierdzenie, że idea doboru utworów oparta w całości na liczbach może być ciekawą alternatywą, z której wielu użytkowników mogłoby realnie korzystać. Rekomendowane użytkownikowi utwory sprawiały wrażenie spójnych w kontekście brzmienia, a nieraz znalezione treści, choć wcześniej słuchaczowi nieznane, były satysfakcjonujące i wpisywały się w zdefiniowany charakter profilu.

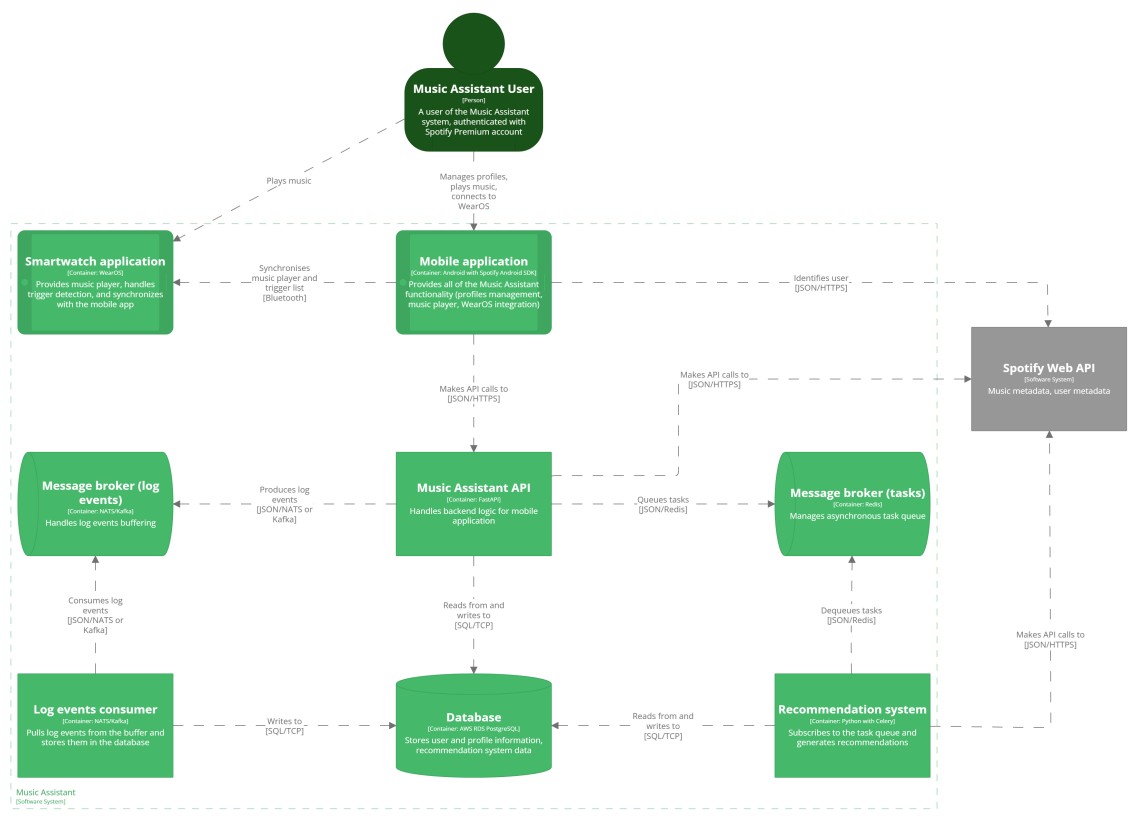

Rysunek 2: Diagram kontenerów Music Assistant

# 4 WNIOSKI

Na podstawie przeprowadzonych testów można stwierdzić, że Music Assistant jest ciekawą alternatywą dla obecnych rozwiązań w branży streamingu muzyki. Rozwiązanie opiera się na numerycznym opisie muzyki dostarczanym przez Spotify API, dzięki czemu aplikacja jest w stanie w ramach profilu oferować użytkownikowi utwory spójne charakterem i o podobnym brzmieniu. Jest to szczególnie ważne dla osób, które chcą mieć wpływ na słuchane treści, jednocześnie nie muszące skupiać się na ich ręcznym doborze.

Algorytm nie jest podatny na popularność rekomendowanych utworów dzięki numerycznej reprezentacji muzyki. Pozwala to odkrywać użytkownikom nowe, mniej znane treści. Ankieta przeprowadzana przy tworzeniu nowego profilu w przyjemny i ciekawy sposób ukierunkowuje profil na konkretny charakter muzyczny, dzięki czemu użytkownik może od początku cieszyć się utworami dostosowanymi do swoich preferencji. Zbieranie danych z sensorów zegarka z WearOS oraz ich interpretacja zgodna ze zdefiniowanymi wyzwalaczami pozwala aplikacji na bieżąco dostosowywać się do kontekstu i sytuacji, przez co Music Assistant ma potencjał stać się nieodłącznym elementem codzienności użytkowników.

Najważniejszym sukcesem projektu jest połączenie wszystkich komponentów i utworzenie działającej i skalowalnej aplikacji, która ze względu na przemyślaną architekturę jest gotowa na wdrożenie produkcyjne oraz na dalszą rozbudowę i ulepszenia, szczególnie w kwestii rekomendacji, sposobu przeprowadzania ankiety oraz użycia danych z sensorów Wear OS.

## 5   KIERUNKI ROZWOJU

Projekt może być rozwijany na wielu płaszczyznach, zarówno komercyjno-biznesowych, jak i funkcjonalnych.

- Najważniejszym i pierwszym kierunkiem rozwoju jest nawiązanie oficjalnej współpracy ze Spotify, aby umożliwić komercjalizację rozwiązania i zapewnić możliwość korzystania z aplikacji przez większą grupę użytkowników.

- Kolejną kwestią jest implementacja aplikacji mobilnej dedykowanej na urządzenia z systemem iOS, co pozwoli dotrzeć do klientów Spotify korzystających z iPhone'ów.

- Ulepszanie i usprawnianie systemu rekomendacji, aby coraz trafniej był w stanie interpretować aktywność użytkownika i dostosowywać parametry profili.

- W aplikacji na urządzenia z systemem Wear OS polem do rozwoju jest ulepszenie interpretacji zdarzeń wykrywanych przez sensory inteligentnego zegarka. Należy powiększyć pulę zdarzeń możliwych do wykrycia przez aplikację oraz zaadresować bardziej złożone sytuacje, w których zdarzenia będą wykrywane.

- Rozważenie implementacji aplikacji na inteligentne zegarki dla innych systemów operacyjnych używanych obecnie przez urządzenia tego typu (np. *watchOS*, *Fitbit OS*, *Garmin OS*). Należy przeanalizować możliwości tych systemów pod kątem zbierania danych z sensorów urządzenia i wykorzystania ich w naszej aplikacji w podobny sposób, jak ma to miejsce teraz na Wear OS.

- Analiza i wykorzystanie konkurencyjnych źródeł informacji o muzyce i serwisów streamingowych, takich jak *YouTube Music*, *Apple Music*, *Tidal*, *SoundCloud*, *Amazon Music* oraz możliwe nawiązanie z nimi współpracy.

## 6   PODZIĘKOWANIA

W tym miejscu chcielibyśmy serdecznie podziękować Pani Doktor Bernadetcie Maleszce, promotorce naszego projektu, za wsparcie merytoryczne i pomoc w kwestiach formalnych związanych z kursem i wydarzeniem ZPI Day. Dziękujemy także Panu Doktorowi Marcinowi Pietranikowi oraz Panu Doktorowi Marcinowi Kawalerowiczowi za wartościowe i praktyczne sugestie. Ogromne podziękowania składamy również Panu Profesorowi Stanisławowi Saganowskiemu za pomoc w określeniu i doprecyzowaniu tematu pracy inżynierskiej oraz za praktyczne sugestie dotyczące rynku i technologii wearable. Dziękujemy też Zespołowi naukowemu Emognition za użyczenie inteligentnego zegarka z systemem Wear OS, który posłużył nam do implementacji i testowania aplikacji na urządzenia ubieralne.

## LITERATURA

[1] Kim Falk. *Praktyczne systemy rekomendacji*. Wydawnictwo Naukowe PWN SA, Warszawa, 2020.

[2] GitLab. Gitlab CI/CD documentation. `https://docs.gitlab.com/ee/ci/`, 2024. [dostęp 23.11.2024].

[3] Spotify. Spotify Android SDK documentation. `https://developer.spotify.com/documentation/android/`, 2024. [dostęp 27.11.2024].

[4] Spotify. Spotify Web API documentation. `https://developer.spotify.com/documentation/web-api/`, 2024. [dostęp 27.11.2024].

[5] Spotify. Understanding recommendations on Spotify. `https://www.spotify.com/us/safetyandprivacy/understanding-recommendations`, 2024. [dostęp 29.11.2024].
