# OpenReview forum: "Music Assistant"
_pwr.edu.pl/Wrocław_University_of_Science_and_Technology/2024/ZPI_Day — Wrocław University of Science and Technology 2024 ZPI Day Submission_

### Official Review · Reviewer_EQXd · 2024-12-03
**Abstrakt został starannie przygotowany pod względem merytorycznym, strukturalnym i językowym**

**Confidence:** 4
**Significance Of Results:** 4
**Overall Quality:** 4

**Compliance With Template:**

5: Very High Quality – The article contains all the required sections, which are written in a very detailed, clear, and error-free manner. The structure is professional and meets expectations, and the content adheres to the highest substantive and formal standards.

**Description Of Results:**

4: High Quality – The results are described in detail and supported by usage examples or evaluations. The description is reliable but may lack full depth of analysis.

**Feedback On Consistency:**

Praca napisana spójnie, choć warto byłoby wzbogacić opis dotyczący architektury systemu oraz algorytmu rekomendacji.

**Potential For Development:**

Wymienione w pracy kierunki dalszych prac wskazują na realistyczne podejście do tematu.

**Project Nature Evaluation:**

Projekt wykazuje cechy pracy inżynierskiej. Opisane aspekty pracy wskazują na realizację założonego celu.

**Technical Language Precision:**

4: High Quality – The language is appropriate for a technical report. Terminology is used correctly, and statements are precise, with only minor shortcomings that do not affect the overall clarity.

---

### Official Review · Reviewer_y4pj · 2024-12-04
**Praca w ciekawy sposób automatyzuje proces doboru utworów muzycznych w platformie Spotify z wykorzystaniem definiowanych profili**

**Confidence:** 5
**Significance Of Results:** 4
**Overall Quality:** 5

**Compliance With Template:**

5: Very High Quality – The article contains all the required sections, which are written in a very detailed, clear, and error-free manner. The structure is professional and meets expectations, and the content adheres to the highest substantive and formal standards.

**Description Of Results:**

5: Very High Quality – The results are described in detail, clearly and comprehensively, supported by thorough evaluation, analysis, and convincing usage examples. The description meets the highest substantive standards.

**Feedback On Consistency:**

Projekt stanowi bardzo ciekawe i w pewnym sensie nowatorskie podejście do problemu. Jedyny problem jaki dostrzegam jest licencja na API Spotify, która - o ile nie zmieniła się w ostatnim czasie - nie zezwala na swobodne wykorzystywanie narzędzia w dowolnym celu. Z tego względu rozwój projektu jest uzależniony od zmian licencyjnych na wykorzystywane narzędzie.
Niemniej jednak wykorzystane narzędzia i metody są dobrane i zastosowane w przemyślany i prawidłowy sposób.

**Potential For Development:**

Projekt - po uwzględnieniu wszystkich ograniczeń licencyjnych może być wprowadzony na rynek komercyjny, gdzie może znaleźć rzeszę odbiorców.

**Project Nature Evaluation:**

Projekt wykazuje dojrzałość i spełnia wszystkie kryteria stawiane projektom inżynierskim. Zastosowane technologie są tożsame ze stosowanymi w rozwiązaniach komercyjnych.

**Technical Language Precision:**

5: Very High Quality – The language is entirely appropriate for a technical report. All terms are used correctly and precisely, and the style is professional, clear, and coherent, without any errors or ambiguities.

---

### Official Review · Reviewer_qmg5 · 2024-12-06
**Recenzja projektu Music Assistant**

**Confidence:** 5
**Significance Of Results:** 4
**Overall Quality:** 5

**Compliance With Template:**

5: Very High Quality – The article contains all the required sections, which are written in a very detailed, clear, and error-free manner. The structure is professional and meets expectations, and the content adheres to the highest substantive and formal standards.

**Description Of Results:**

5: Very High Quality – The results are described in detail, clearly and comprehensively, supported by thorough evaluation, analysis, and convincing usage examples. The description meets the highest substantive standards.

**Feedback On Consistency:**

Abstrakt napisany jest w sposób spójny i logiczny. Sugerowałabym również zdecydować się na jedną formę (najlepiej bezosobową). W niektórych miejscach autorzy piszą w formie bezosobowej, natomiast w innych stosują 1 os. liczby mnogiej.

**Potential For Development:**

Autorzy zidentyfikowali szereg możliwości rozwoju, zarówno związanych z funkcjonalnością aplikacji, jak i z rozwojem biznesowym. Wydaje mi się, że przed potencjalnym wdrożeniem i/lub rozwojem, należałoby przeprowadzić badania wskazujące potencjalne zainteresowanie zaprojektowanym rozwiązaniem.

**Project Nature Evaluation:**

Układ pracy jest właściwy. Wydaje mi się, że cel powinien był bardziej uwypuklony. Cel wynika z treści, natomiast brakuje mi większej precyzji w jego wskazaniu. Zastosowano odpowiednie narzędzia i technologie. Uważam, że projekt spełnia wymogi stawiane projektom inżynierskim.

**Technical Language Precision:**

5: Very High Quality – The language is entirely appropriate for a technical report. All terms are used correctly and precisely, and the style is professional, clear, and coherent, without any errors or ambiguities.

---

### Decision · Program_Chairs · 2024-12-10

Accept (Oral)